# Robust Multi-Organ Nucleus Segmentation Using a Locally Rotation Invariant Bispectral U-Net

**Valentin Oreiller**[1,2]                                               valentin.oreiller@hevs.ch
**Julien Fageot**[3]                                                      julien.fageot@epfl.ch
**Vincent Andrearczyk**[1]                                      Vincent.Andrearczyk@hevs.ch
**John O. Prior**[2]                                                        John.Prior@chuv.ch
**Adrien Depeursinge**[1,2]                                 Adrien.Depeursinge@hevs.ch

[1] *Institute of Information Systems, HES-SO, Sierre, Switzerland*

[2] *Service of Nuclear Medicine and Molecular Imaging, CHUV, Lausanne, Switzerland*

[3] *AudioVisual CommunicationS Laboratory (LCAV), EPFL, Lausanne, Switzerland*

**Editors:** Under Review for MIDL 2022

## Abstract

Locally Rotation Invariant (LRI) operators have shown great potential to robustly identify biomedical textures where discriminative patterns appear at random positions and orientations. We build LRI operators through the local projection of the image on circular harmonics followed by the computation of the bispectrum, which is LRI by design. This formulation allows to avoid the discretization of the orientations and does not require any criterion to locally align the descriptors. This operator is used in a convolutional layer resulting in LRI Convolutional Neural Networks (LRI CNN). To evaluate the relevance of this approach, we used it to segment cellular nuclei in histopathological images. We compared the proposed bispectral LRI layer against a standard convolutional layer in a U-Net architecture. While they performed equally in terms of F-score, the LRI CNN provided more robust segmentation with respect to orientation, even when rotational data augmentation was used. This robustness is essential when the relevant pattern may vary in orientation, which is often the case in medical images.

**Keywords:** Local Rotation Invariance, Convolutional Network, Deep Learning, Segmentation, Bispectrum

## 1. Introduction

Robustness of Convolutional Neural Networks (CNNs) to changes in orientations of the input structures (*e.g.* nucleus, glands) has been little investigated and may have an important impact on the usability of the methods in practice. Biomedical textures are composed of local patterns that appear at random positions and orientations. Local Rotation Invariant (LRI) operators have been shown to be crucial to characterize such texture (Depeursinge et al., 2018). A common strategy to design LRI operators is to align local descriptors. This includes the Maximum Response 8 (MR8) filterbank (Varma and Zisserman, 2005) and Local Binary Patterns (LBP) (Ojala et al., 2002; Ahonen et al., 2009). Other methods relying on steerability have been proposed to avoid error due to orientation sampling, such as steerable filters (Unser and Chenouard, 2013; Zhao and Blu, 2020; Fageot et al., 2021), Riesz (Dicente Cid et al., 2017), and steerable wavelets (Depeursinge et al., 2017; Puspoki et al., 2019). Another well known method is the scale-invariant feature transform (Lowe,

2004). These methods typically require discretizing orientations and an arbitrary criterion to select the dominant local orientation on which the descriptor is aligned. Built-in LRI approaches have been proposed to avoid using such arbitrary criterion. For instance, the power spectrum was used in (Depeursinge et al., 2018; Andrearczyk et al., 2019) which allows obtaining a LRI operator continuously defined on the rotation domain. In this work, we design a LRI operator based on a similar but more evolved descriptor, *i.e.* the bispectrum, that can be embedded in a CNN.

CNNs have revolutionized the field of computer vision and biomedical image analysis. Rotation invariance in CNNs is still mainly induced via data augmentation either at training- or test-time. However, built-in rotation equivariance was shown to outperform both training- and test-time augmentation on histopathological image analysis (Lafarge et al., 2021). Rotation-equivariant networks also showed improved robustness to geometric adversarial perturbations (Dumont et al., 2018). A large body of research has focused on designing networks with built-in rotation equivariance and are mainly based on discretized rotations (*i.e.* group equivariant CNN) or steerable filters (Weiler et al., 2018, 2017; Cohen and Welling, 2016; Bekkers, 2019; Kondor and Trivedi, 2018; Cohen et al., 2019).

In this work, we propose a CNN design that is invariant to local rotations rather than rotation equivariant. The motivation is that such a design will provide substantial robustness to changes in pattern orientation when identifying biomedical textures. Furthermore, the proposed LRI operators are also globally rotation equivariant (Andrearczyk et al., 2020). While global rotation invariance may be obtained by data augmentation, invariance to local rotation can not be achieved in this way. In (Andrearczyk et al., 2020), the authors proposed two different designs to implement LRI CNNs, one with steerable filters and another one based on the power spectrum. In (Eickenberg et al., 2017), the power spectrum was used within the scattering transform framework (Ablowitz et al., 1974) to obtain global rotation equivariant feature maps. Those works closely relate to our design. One key difference is that we use the bispectrum rather than the power spectrum. Our design relies on the shift invariance property of the bispectrum which translates into rotational invariance for functions defined on the circle. We chose the bispectrum over the power spectrum for its completeness, *i.e.* the bispectrum completely characterizes a function up to a shift (Kakarala, 2012). We evaluated this design with a simple U-Net (Ronneberger et al., 2015) on histopathological images. The evaluation were greatly inspired by the work of Lafarge et al. (2021), in order to have an external comparison. However, our results are not directly comparable since we did not use the same training/testing/validation split.

## 2. Methods

In this section, we develop the theoretical background as well as the implementation details to design a LRI CNN. The main idea is to obtain a LRI convolutional layer that is functionally identical to a standard convolutional layer. Then, we evaluate our layer in a CNN and compare it to a CNN with the same architecture but with standard layers. This work focuses on developing 2D CNNs that are invariant to the orientation at which local patterns appear. The proposed design relies on the rotation invariance property of the bispectrum.

## 2.1. Notations

We consider 2D images as functions $I \in L_2(\mathbb{R}^2)$, where the value $I(\boldsymbol{x}) \in \mathbb{R}$ corresponds to the pixel intensity at location $\boldsymbol{x} = (x_1, x_2) \in \mathbb{R}^2$. The rotation of an image $I$ is written as $I(\mathrm{R}\cdot)$, where $\mathrm{R} \in SO(2)$ is the corresponding 2D rotation matrix.

The circle is denoted as $\mathbb{S}^1 = \{\boldsymbol{x} \in \mathbb{R}^2 : ||\boldsymbol{x}||_2 = 1\}$. Polar coordinates are defined as $(x_1, x_2) = (\rho \cos(\theta), \rho \sin(\theta))$ with $\rho \geq 0$ and $\theta \in [0, 2\pi)$. We use the following notation for the mapping from polar to cartesian: $\rho(\boldsymbol{x}) = ||\boldsymbol{x}||$ and $\theta(\boldsymbol{x})$ the standard mapping from $\boldsymbol{x}$ to its polar angle. For clarity purposes, we often do not disclose the dependency on $\boldsymbol{x}$ for $\rho$ and $\theta$. We consider square-integrable functions defined on the circle $f \in L_2(\mathbb{S}^1)$ and express them as functions of the polar angle $f(\theta)$. The inner product is defined by $\langle f, g \rangle_{L_2(\mathbb{S}^1)} = \int_0^{2\pi} f(\theta)\overline{g(\theta)}\mathrm{d}\theta$. The rotation of a function $f$ by an angle $\theta_0$ is simply the function "shifted" by that angle $i.e.$ $f(\cdot - \theta_0)$. The Fourier transform of the function $f$ is defined as $\hat{f}[n] = \int_0^{2\pi} f(\theta)e^{-jn\theta}\mathrm{d}\theta$ for any $n \in \mathbb{Z}$. The triangle function is referred to as $x \in \mathbb{R} \mapsto \mathrm{tri}(x)$ and is defined as $\mathrm{tri}(x) = 1 - |x|$ if $|x| < 1$ and $\mathrm{tri}(x) = 0$ otherwise.

## 2.2. LRI Operators

Our mathematical formalism relies on the concepts of image operators acting over continuous images, as presented in (Depeursinge and Fageot, 2017). This work focuses on image operators $\mathcal{G}$ that are LRI as previously introduced in (Andrearczyk et al., 2020). An operator $\mathcal{G}$ is LRI if it satisfies the three following properties:

- *Locality*: there exists $\rho_0 > 0$ such that, for every $\boldsymbol{x} \in \mathbb{R}^2$ and every image $I \in L_2(\mathbb{R}^2)$, the quantity $\mathcal{G}\{I\}(\boldsymbol{x})$ only depends on local image values $I(\boldsymbol{y})$ for $\|\boldsymbol{y} - \boldsymbol{x}\| \leq \rho_0$.

- *Global equivariance to translations:* For any $I \in L_2(\mathbb{R}^2)$,

$$\mathcal{G}\{I(\cdot - \boldsymbol{x}_0)\} = \mathcal{G}\{I\}(\cdot - \boldsymbol{x}_0) \quad \text{for any } \boldsymbol{x}_0 \in \mathbb{R}^2.$$

- *Global equivariance to rotations:* For any $I \in L_2(\mathbb{R}^2)$,

$$\mathcal{G}\{I(\mathrm{R}_0\cdot)\} = \mathcal{G}\{I\}(\mathrm{R}_0\cdot) \quad \text{for any } \mathrm{R}_0 \in SO(2).$$

The simplest example of a LRI operator is the convolution with filter

$$\mathcal{G}\{I\}(\boldsymbol{x}) = (I * h)(\boldsymbol{x}), \tag{1}$$

where $h$ is isotropic with finite support, $i.e.$ $h(x_1, x_2) = h(\rho)$ is purely radial and vanishes for $\rho > \rho_0$ for some fixed $\rho > 0$ (Andrearczyk et al., 2020, Prop. 1). However, isotropic filters are limited since they discard local directional information. We will now see how we can extend this notion of equivariance to directional sensitive operators using the bispectrum.

## 2.3. Bispectral LRI Operators and Layers

We introduce the theoretical background and methodology to implement our bispectral image operators. They have the advantage of being LRI and sensitive to directional information. We also detail how these operators can be embedded into a convolutional layer.

### 2.3.1. The Bispectrum: A Complete Set of Rotation Invariant Features

We first focus on features $\mathcal{H} : L_2(\mathbb{S}^1) \to \mathbb{R}$ of circular functions that are rotation invariant, *i.e.* such that

$$\mathcal{H}\{f(\theta)\} = \mathcal{H}\{f(\theta - \theta_0)\} \tag{2}$$

for any function $f$ and any angle $\theta_0 \in [0, 2\pi)$. The typical example is the power spectrum of $f$ defined from its Fourier series coefficients $\hat{f}[n]$. For any fixed discrete frequency $n_0 \in \mathbb{Z}$, the Fourier feature $f \mapsto |\hat{f}[n_0]|^2$ is easily shown to be rotation invariant in the sense of (2). However, the power spectrum discards the phase information of $\hat{f}$ and thus, does not allow for the complete characterization of a polar function.

For this reason, we consider a more elaborated Fourier-based feature, named the bispectrum, that retains the rotation invariance while keeping the phase information (Bartelt et al., 1984; Kakarala, 2012). The bispectrum of a function $f \colon L^2(\mathbb{S}^1) \to \mathbb{R}$ is defined for any $n_1, n_2 \in \mathbb{Z}$ as

$$b_f[n, n'] = \hat{f}[n]\hat{f}[n']\overline{\hat{f}[n + n']}. \tag{3}$$

One readily verifies that, for any $n, n' \in \mathbb{Z}$, the feature $f \mapsto b_f[n, n']$ is rotation invariant. The bispectrum is *complete* (Kakarala, 2012) in the sense that it discriminates between two functions up to a rotation which motivates our choice of using it over the power spectrum to design LRI layers.

### 2.3.2. Bispectral LRI Operators

The bispectrum is defined for circular functions in $L^2(\mathbb{S}^1)$ and can be used to build image operators. In this section, we fix a radial function $h(\rho) \in L_2(\mathbb{R}^2)$ and consider the steerable kernel $\kappa_n(\rho, \theta) = h(\rho)e^{\mathrm{j}n\theta}$ associated to the discrete frequency $n \in \mathbb{Z}$. We moreover introduce the convolution operator $\mathcal{C}_n\{I\}(\boldsymbol{x}) = (I * \kappa_n)(\boldsymbol{x})$. We observe that we can write, for any fixed position $\boldsymbol{x}_0 \in \mathbb{R}^2$,

$$\mathcal{C}_n\{I\}(\boldsymbol{x}_0) = \int_0^{2\pi} \left( \int_0^{+\infty} \left(I(\boldsymbol{x}_0 - \cdot)\right)(\rho, \theta)h(\rho)\rho\mathrm{d}\rho \right) e^{-\mathrm{j}n\theta}\mathrm{d}\theta.$$

We can interpret the circular function $\theta \mapsto I^h_{\boldsymbol{x}_0}(\theta) := \int_0^{+\infty} \left(I(\boldsymbol{x}_0 - \cdot)\left(\rho, \theta\right)h(\rho)\rho\mathrm{d}\rho$ as the radial projection of the shifted image $I(\boldsymbol{x}_0 - \cdot)$ against the radial profile $h$. Hence, $\mathcal{C}_n\{I\}(\boldsymbol{x}_0)$ performs the $n^{\text{th}}$ Fourier coefficient of the periodic function $I^h_{\boldsymbol{x}_0}$.

For any $n, n' \in \mathbb{Z}$, we define the image operator $\mathcal{G}_{n,n'}$ as

$$\mathcal{G}_{n,n'}\{I\}(\boldsymbol{x}) = \mathcal{C}_n\{I\}(\boldsymbol{x})\mathcal{C}_{n'}\{I\}(\boldsymbol{x})\overline{\mathcal{C}_{n+n'}\{I\}(\boldsymbol{x})}. \tag{4}$$

Then, we see by comparing (4) with (3) that, for any fixed position $\boldsymbol{x}_0$, $\mathcal{G}_{n,n'}\{I\}(\boldsymbol{x}_0)$ is the bispectrum of the projection $I^h_{\boldsymbol{x}_0} \in L_2(\mathbb{S}^1)$. We call $\mathcal{G}_{n,n'}$ the *bispectral operator of frequencies $n, n'$*. Its main invariance properties are summarized in Theorem 1, whose proof is provided in Appendix A.

**Theorem 1** *Let $n, n' \in \mathbb{Z}$ and $\rho \mapsto h(\rho)$ a radial profile with finite support. Then, the bispectrum operator $\mathcal{G}_{n,n'}$ is LRI.*

### 2.3.3. Implementation of the LRI Layer

We fix a maximal order $N \geq 0$ and consider the bispectral operators $\mathcal{G}_{n,n'}$ for any $n, n' \geq 0$ such that $n + n' \leq N$. By doing so, we only consider angular frequencies smaller than $N$. Moreover, we only compute non-repeating pairs as $\mathcal{G}_{n,n'} = \mathcal{G}_{n',n}$. We define $L$ as the number of combinations of $n$, $n'$ such that $n \leq n'$ and $n + n' \leq N$.

For any $n, n'$, the application of the bispectral operator $\mathcal{G}_{n,n'}$ at each layer of the bispectral LRI network is implemented in four steps as depicted in Fig. 1. First, the feature maps are computed as a complex convolution $\mathcal{C}_n^o(\boldsymbol{x}_0) = \sum_{i=1}^{C}(y_i(\boldsymbol{x}) * h_n^{i,o}(\boldsymbol{x})e^{-\mathrm{j}n\theta(\boldsymbol{x})})(\boldsymbol{x}_0)$, with $y_i$ the $i^{\text{th}}$ channel of the previous feature maps, $h_n^{i,o}$ the filters that are learned by gradient descent. The parametrization of the filters $h_n^{i,o}$ is detailed in Section 2.3.4. The indices $i$ and $o$ run through $[1, \ldots, C_{in}]$ and $[1, \ldots, C_{out}]$ respectively and represent the input and output channels of the layer.

The second step consists in applying (4) to the feature maps $\mathcal{C}_n^o(\boldsymbol{x})$, yielding the desired operator $\mathcal{G}_{n,n'}$.

The third step is the concatenation of the real and imaginary part of $\mathcal{G}_{n,n'}$, which is followed by a point-wise non-linearity of the following form $\sigma(x) = \mathrm{sign}(x)\log(1+|x|)$. This choice is made to avoid vanishing and exploding gradients with the cubic nature of the bispectral feature maps (see Eq. (4)). Learned biases are added to the resulting feature maps and Rectified Linear Unit (ReLU) is applied.

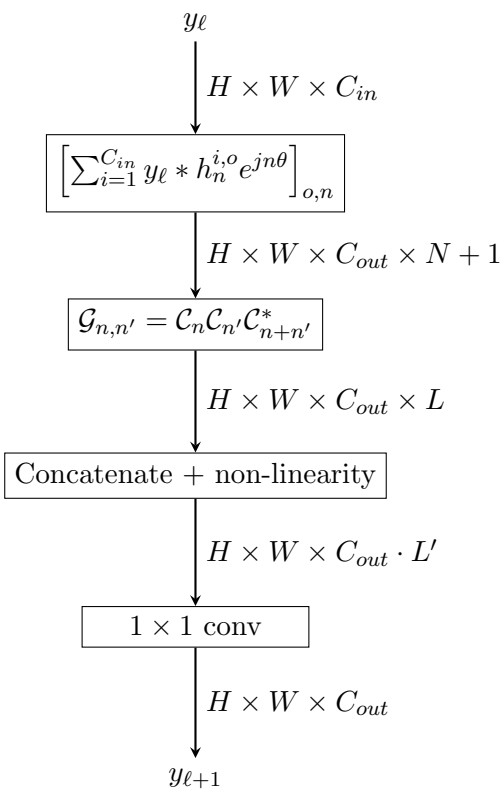

Figure 1: The proposed bispectrum-based LRI convolutional layer.

In the last step, the number of features maps is reduced by a $1 \times 1$ convolution to obtain $C_{out}$ output channels. This whole process results in a layer that takes as an input $C_{in}$ feature maps and outputs $C_{out}$ feature maps like a standard convolutional layer.

### 2.3.4. Parametrization of the Radial Profiles

The radial profiles $h_n^{i,o}$ are parametrized as follows:

$$h_n^{i,o}(\rho) = \sum_{j=0}^{J} w_{n,j}^{i,o} \psi_j(\rho), \tag{5}$$

where the $w_{n,j}^{i,o}$ are the learnable parameters of the layer, $i \in [1, \cdots, C_{in}]$, $o \in [1, \cdots, C_{out}]$ and $0 \leq n \leq N$. The radial functions $\psi_j$ are chosen as $\psi_j(\rho) = \mathrm{tri}(\rho - j)$.

### 2.4. Dataset

We tested our design on a subset of the dataset proposed in the MoNuSeg 2018 challenge (Kumar et al., 2019) which consists of 24 Hematoxylin and Eosin (H&E) stained images selected from whole slice images acquired at the commonly used $40\times$ magnification provided by The Cancer Genome Atlas (Koboldt et al., 2012). This subset contains 6 $1000 \times 1000$ images per tissue type for a total of four different tissue types (breast, liver, kidney, and prostate). Nuclei instance segmentation is available for these 24 images. We followed a similar splitting scheme than proposed by (Lafarge et al., 2019), namely $4 \times 3$ images for training, $4 \times 1$ for validation and $4 \times 2$ for testing. We repeated ten random splits to evaluate the variation of the models. As preprocessing, we used the method described in (Macenko et al., 2009) to normalize the H&E images. We adapted the code from https://github.com/schaugf/HEnorm_python to fit our needs.

### 2.5. Network Architecture and Training

The networks used in this work were based on the U-Net architecture (Ronneberger et al., 2015). However, we used a lighter version of the U-Net, as proposed in (Lafarge et al., 2021), which contains only two levels of down-sampling. All the convolutional layers have a kernel size of $5 \times 5$ and are connected to a batch normalization followed by a ReLU. The encoder path includes three convolutions layers with max-pooling to reduce the spatial dimension. The number of feature maps for each layer respectively are 8, 16, and 32. The decoder path contains 2 layers preceded by a bi-linear upsampling. The final prediction is modeled as a three classes probability, *i.e.* nucleus core, nucleus border, and background. The prediction is computed with a $1 \times 1$ convolution with a softmax activation. The final output is post-processed to obtain instance segmentation of each nucleus. This post-processing consists in binarizing the prediction with a threshold of 0.5, then the core and border prediction are respectively used as seed and landscape for a watershed algorithm[1] (Falcão et al., 2004).

The networks were trained by minimizing the class-balanced cross-entropy with an Adam optimizer and a learning rate of $10^{-3}$. The models were trained on patches of $60 \times 60$ randomly drawn from the training set with a batch size of 16. We applied multiple of $90°$ rotation as data augmentation[2] as well as random brightness shift. The training was run for a maximum number of epochs of 200 and we applied an early stop monitoring the F-score. The experiments were performed on an Nvidia Tesla K80. The code for the implementation is available on our GitHub repository[3].

### 2.6. Metrics and Evaluation

Two types of experiments were conducted. We first evaluated the performance of our bispectral LRI U-Net against a standard U-Net. The metric used for this experiment is the F-score and we considered a match when the predicted nucleus had more than 50% overlap with the ground-truth nucleus. Since the radial profiles of the proposed LRI layer do not cover the entire $5 \times 5$ kernel, we used masked kernels in the standard U-Net to ensure a

---

1. docs.scipy.org/doc/scipy/reference/generated/scipy.ndimage.watershed_ift.html, March 2022.
2. We also evaluated the training without this augmentation, but did not observe any significant changes.
3. github.com/voreille/2d_bispectrum_cnn, March 2022.

fair comparison. The masked kernel consists of removing the four pixels in the corners. All models were trained and tested on the same 10 train/validation/testing splits to assess performance variation.

In the second experiment, we evaluated the robustness of the predictions made by the two designs in terms of 90°-rotation equivariance. We fed the networks with the same image rotated at different orientations. Then, we applied the inverse rotation to the output maps and compare the difference in pixel-wise prediction. We used the Root Mean Square Error (RMSE) on the three classes of raw probabilities prediction as well as the Dice Similarity Coefficient (DSC) on the post-processed probability to measure the overlap of predictions for each orientation, indicating the robustness of prediction with respect to input orientation.

## 3. Results

Fig. 3 relates the F-score for varying maximum degree $N$ of the bispectral U-Net and standard U-Net. The best performing bispectral U-Net had an F-score of $0.7157 \pm 0.0328$ with a maximum degree $N = 7$ and 136,147 parameters (standard U-Net 71,571 parameters). Thus, we trained a standard U-Net with more feature maps to increase the number of parameters (134,709 parameters) which obtained an average F-score of $0.7324 \pm 0.0326$.

To account for the discrepancy in the number of parameters at different maximal degrees $N$, we trained a bispectral U-Net with $N = 0$ and more feature maps to obtain a network of 45,779 parameters (comparable to the number of parameters of the network with N=3). The resulting network achieved an average F-score of $0.6592 \pm 0.0286$.

Table 1 summarizes the average RMSE and DSC between predictions of rotated images. The average was calculated on all the images from the testing set of one split and all pairs of 90° rotations. Fig. 2 illustrates the pixel-wise variation between predictions when fed to the networks at different orientations.

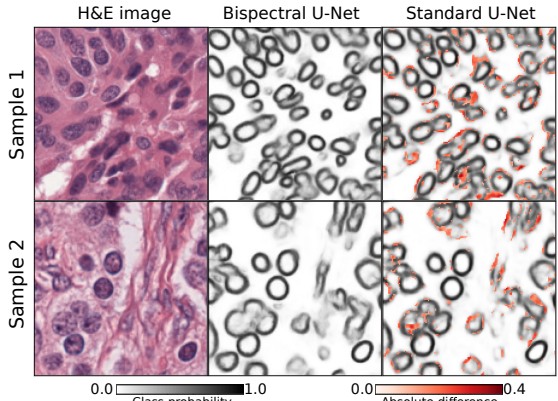

Figure 2: Illustration of the prediction robustness with respect to input orientation. The middle and right columns depict the probability prediction of the nucleus border class for the bispectral and standard U-Nets, respectively. The red color map indicates the mean pixel-wise differences averaged across the six pairs of 90° rotations. These differences are almost null for the bispectral U-Net.

## 4. Discussions and Conclusion

We proposed a novel 2D LRI layer based on the bispectrum that can be integrated into any CNN architecture. This design aims to improve the robustness of the predictions when

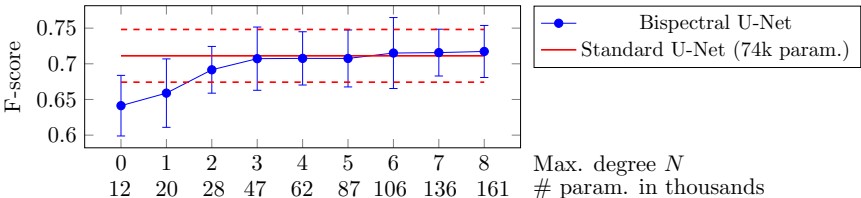

Figure 3: Performance of the different networks. Average F-scores are reported across ten repetitions of the proposed bispectral U-Net evaluated at different maximum degrees $N$ (blue) and standard U-Net (red). Error bars and dashed lines indicate the standard deviation.

Table 1: Quantitative evaluation of segmentation robustness. It is worth noting that rotational data augmentation was used during training for both approaches.

| Model | RMSE border | RMSE core | RMSE background | DSC |
|---|---|---|---|---|
| Standard U-Net | $8.49 \pm 1.35$ % | $7.66 \pm 1.72$ % | $9.15 \pm 1.43$ % | $0.9153 \pm 0.0205$ |
| Bispectral U-Net ($N = 7$) | 2.50e-5 $\pm$0.78e-5 % | 2.26e-5 $\pm$0.77e-5 % | 2.53e-5 $\pm$0.87e-5 % | $0.9876 \pm 0.0044$ |

inputs do not have a standardized orientation, or when local structures of interest can appear at any orientation.

We first presented the bispectral operators and demonstrated their LRI property in Section 2.3.2 and Appendix A. We also detailed how to use them in a convolutional layer in Section 2.3.3. Second, we incorporated the LRI layer into a U-Net to allow robust segmentation of multi-organ nuclei in histopathology images. We observed that the segmentation performance of the LRI U-Net is on par with a standard U-Net (see Fig. 3). While the bispectral U-Net was slightly outperformed by the standard U-Net, it is difficult to evaluate to which extent the post-processing had a role in this difference (see Appendix B).

However, an important gain was obtained in terms of robustness with respect to the orientation of the input (see Table 1) thanks to the rotation equivariance property of the used image operators. This robustness is crucial for most medical image analysis tasks where structures of interest often appear at various orientations. We observed that standard methods lack robustness, even when rotational data augmentation is used (see Table 1 and Fig. 2). While most studies focused on classification or segmentation performance alone, robustness to changes in input orientation was little investigated and may have important consequences on the usability of the models.

Our work recognizes several limitations. The segmentation performance presented in Fig. 3 is not at the level of the state of the art on this dataset. Our goal was to compare with standard baseline methods such as the U-Net without using refinements *e.g.* postprocessing of the segmentation maps or ensembling. Future work includes the extension of the bispectral operator to 3D and extensive comparisons with group-equivariant approaches.

## Acknowledgments

This work was partially supported by the SNSF grants 205320_179069 and P400P2_194364, the SPHN IMAGINE project, and the Hasler Foundation.

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

## Appendix A. Proof of Theorem 1

For any $n \in \mathbb{Z}$, the operator $I \mapsto \mathcal{C}_n\{I\} = I * \kappa_n$ is equivariant to translations (as a convolution) and local (due to the fact that $h$, and therefore $\kappa_n(\rho, \theta) = h(\rho)e^{jn\theta}$, have a finite support). Then, the operator $\mathcal{G}_{n,n'}$ inherits these properties, as is clear from its definition in Eq. (4).

We moreover observe that, for any rotation matrix $R_{\theta_0}$ associated with the angle $\theta_0$, we have that

$$I(R_{\theta_0}\cdot) * \kappa_n = (I * \kappa_n(R_{\theta_0}^{-1}\cdot))(R_{\theta_0}\cdot) = e^{jn\theta_0}(I * \kappa_n)(R_{\theta_0}\cdot), \qquad (6)$$

where the first inequality comes from the relation $f(R_{\theta_0}) * g = (f * g(R_{\theta_0}^{-1}\cdot))(R_{\theta_0}\cdot)$ valid for any $f, g \in L_2(\mathbb{R}^2)$ and the second uses that $\kappa_n(R_{\theta_0}\cdot) = e^{jn\theta_0}\kappa_n$. This implies that, for any image $I$ and any $\boldsymbol{x} \in \mathbb{R}^2$,

$$\mathcal{C}_n\{I(R_{\theta_0}\cdot)\} = I * (\kappa_n(R_{\theta_0}\cdot))(\boldsymbol{x}) = e^{jn\theta_0}\mathcal{C}_n\{I\}(R_{\theta_0}\boldsymbol{x}). \qquad (7)$$

We have therefore that

$$\mathcal{G}_{n,n'}\{I(R_{\theta_0}\cdot)\}(\boldsymbol{x}) = e^{jn\theta_0}e^{jn'\theta_0}\overline{e^{j(n+n')\theta_0}}\mathcal{C}_n\{I\}(R_{\theta_0}\boldsymbol{x})\mathcal{C}_{n'}\{I\}(R_{\theta_0}\boldsymbol{x})\mathcal{C}_{n+n'}\{I\}(R_{\theta_0}\boldsymbol{x})$$
$$= \mathcal{G}_{n,n'}\{I\}(R_{\theta_0}\boldsymbol{x})$$

and the operator $\mathcal{G}_{n,n'}$ is globally rotation equivariant. Being local and equivariant to shifts and rotations, $\mathcal{G}_{n,n'}$ is LRI.

## Appendix B. Additional Results

In this Appendix, we report additional results. First, we computed additional metrics to investigate the performance difference between the bispectral and the standard U-Nets. The precision and recall for the bispectral U-Net (N=7) were $0.7004 \pm 0.0617$ and $0.7500 \pm 0.0350$, respectively. For the standard U-Net, we obtained $0.7156 \pm 0.0505$ and $0.7686 \pm 0.0400$. We also trained a standard U-Net without masking the kernels, obtaining an F-score of $0.7318 \pm 0.0220$.

Second, we compare the computational time between the two approaches. The average forward time on 80 1000x1000 images for the bispectral U-Net with $N = 0, 2, 4$, and 7 were 0.14, 0.42, 1.05, and 2.17 seconds respectively. The average forward time for the standard U-Net was 0.07 seconds.

Finally, we report some examples of predictions made by the bispectral and standard U-Net. Fig. 4 illustrates the predictions on a patch where the bispectral U-Net outperforms the standard U-Net. Fig. 5 shows a patch where the bispectral U-Net over segments and thus performs worse than the spectral U-Net.

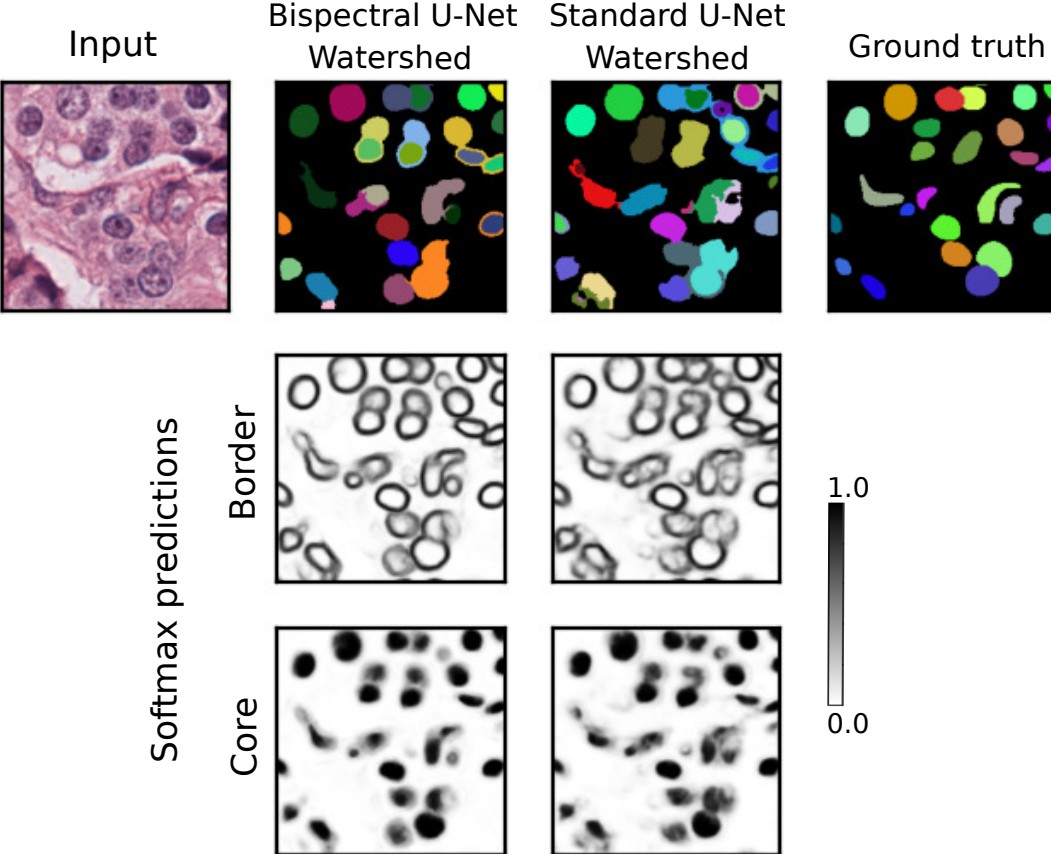

Figure 4: Illustration of predictions where the bispectral U-Net outperforms the standard U-Net. The F-score on this patch for the bispectral and standard U-Net are, respectively, 0.8929 and 0.7931. The colors in the top row images are used to highlight the different nucleus instances obtained after the application of the watershed algorithm.

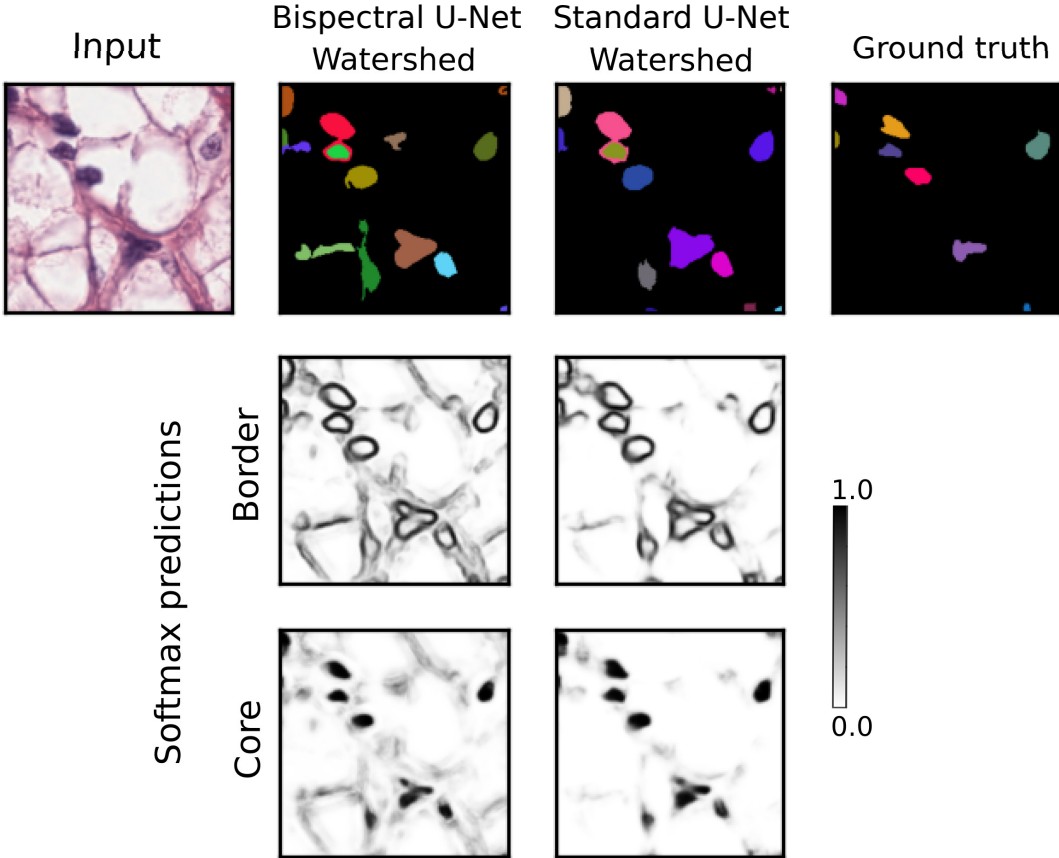

Figure 5: Illustration of predictions where the bispectral U-Net over segments. The F-score on this patch for the bispectral and standard U-Net are, respectively, 0.6667 and 0.7619. The colors in the top row images are used to highlight the different nucleus instances obtained after the application of the watershed algorithm.

The quantitative results seem to indicate that the bispectral U-Net always performed a little worse than the standard U-Net. However, as highlighted in Fig. 4 and 5, it is difficult to assess whether these differences come from the post-processing step.

## Appendix C. Comparison with Spectral U-Net

This appendix describes the results with a similar architecture to the bispectral U-Net. However, the invariant used here is the spectrum yielding a spectral U-Net, which is very similar to the design proposed in (Eickenberg et al., 2017) or in (Andrearczyk et al., 2020). The design of the LRI layer is almost the same and the associated LRI operator is defined as:

$$\mathcal{G}_n\{I\}(\boldsymbol{x}) = \mathcal{C}_n\{I\}(\boldsymbol{x})\overline{\mathcal{C}_n\{I\}(\boldsymbol{x})}. \tag{8}$$

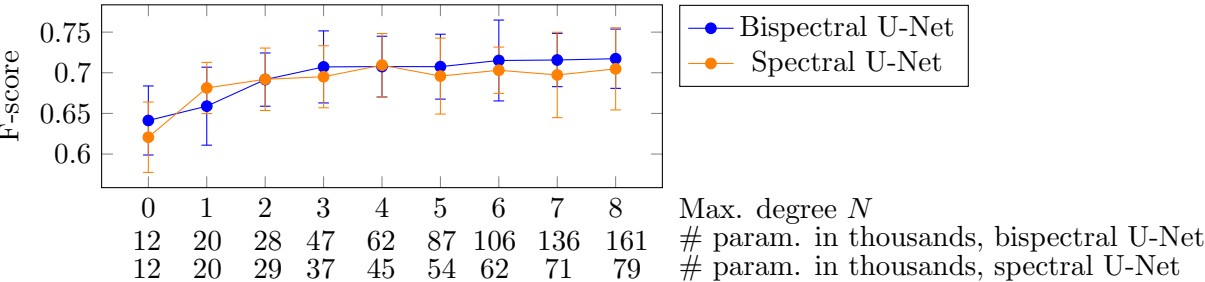

Figure 6: Performance evaluation of the different networks. The average F-scores across 10 repetitions of the proposed bispectral (blue) and spectral (orange) U-Net evaluated at different maximum degrees $N$ are reported.

This invariant is equivalent to taking the modulus, *i.e.* spectrum, of the Fourier coefficients $\mathcal{C}_n$. The results are reported in Fig. 6.

The results suggest slightly better performance for the bispectral U-Net, but the difference remains too marginal to conclude.

