# OpenReview forum: "Robust Multi-Organ Nucleus Segmentation Using a Locally Rotation Invariant Bispectral U-Net"
_MIDL.io/2022/Conference — MIDL 2022_

### Official Review · Reviewer_7L6f · 2022-01-05

**Confidence:** 5
**Preliminary Rating:** 4
**Recommendation:** Oral

**Summary:**

The main contribution is a local rotation invariant layer that can be combined with any network (the presented approach itself demonstrates integration with a Unet) to achieve local rotation invariance in segmentation. The paper is very well-written and the methods are clearly presented. The use of open-source dataset for benchmarking performance is also good.

**Strengths:**

A very well-written paper with clearly presented methods and results. The developed method is novel and may provide increased robustness to rotational differences in the images. Discussion is also well-written with the limitations of the method presented. However, the paper could be improved by explaining why the method underperforms would be good. Some analysis related to using LRI with the standard Unet and the relation to increasing F-score with the parameters is presented -- this could be expanded a bit more in the discussion).

**Weaknesses:**

Some discussion of computational demands of using LRI should be presented. To what extent does the computational demands increase and how does that compare to the accuracy gains.
While the authors acknowledge poor performance of the proposed approach, the reasons of underperformance should be discussed and elaborated more.
The robustness experiment could be expanded a bit more or at least explained a bit better to show how the robustness increases with increasing rotations. The use of rotation in the data augmentation is curious if the method is supposed to be rotation equivariant. This should be better explained.

**Deanonymize Review:**

no

**Detailed Comments:**

In general, the paper is very well written. The methodological contributions and improvement with respect to prior work is clearly presented. The paper would benefit by presenting a few more visual example results - the example Figure 2, shows absolute difference for Unet but not for the proposed method or is it a visual artifact. Also recommend using a different color contrast for the class probability map if possible. If there are any misclassifications in either method, that is not visible.

Please see my comments in weakness for further comments.

**Paper Type:**

methodological development

**Questions To Address In The Rebuttal:**

What is the computational tradeoff with respect to the accuracy gain using this approach? Clearly the presented results do not show accuracy gain - but what is the added computational cost?

The robustness experiment could be expanded or explained a bit better to show how the robustness improves with increasing rotations.

**Special Issue:**

no

---

### Official Review · Reviewer_MueZ · 2022-01-25

**Confidence:** 3
**Preliminary Rating:** 3
**Recommendation:** Poster

**Summary:**

The paper proposes a novel neural network layer based on the bispectrum that is locally rotation invariant, a so-called LRI layer. A small U-net is compared to the same architecture where all convolution layers are substituted with LRI layers (LRI CNN) on histopathological images. Considering the F-score, the LRI CNN performed worse while having more parameters. However, the experiments show that LRI CNNs maintain a better equivariance towards 90° rotations.
The presented idea is good, and the paper is well-crafted. Nevertheless, the benefits of LRI and overall motivations could be explained better. Furthermore, the experiments may be too few and simple to justify a full paper.

**Strengths:**

The authors present a novel approach to obtain local rotation invariance based on the bispectrum and show that the proposed LRI CNNs are robust against 90°-rotations. The paper is well written and has a substantial methods section that provides a theoretical explanation for why the bispectrum is LRI. Moreover, the concept of LRI is based on a large body of previous work.

**Weaknesses:**

Arguably, using the bispectrum is an intriguing and novel approach. Nevertheless, the paper fails to really convince that using the proposed LRI-CNN is a feasible alternative to standard CNNs in biomedical segmentation. The introduction seems relatively short to a reader not familiar with LRI operators. From reading the paper alone, it is hard to tell why LRI operators are well suited for biomedical data. It would be helpful to understand what features is the LRI-layer supposed to extract and why would such behavior be beneficial?

Furthermore, the related work and experiments suggest that rotation equivariance (RE) is a far more desirable trait since most cited deep learning approaches focus on RE, and so do the presented experiments.

The main weakness of the paper is that there are only two experiments with a relatively small U-net and only one dataset. Additionally, the paper lacks comparison with other RE- or LRI approaches. Since the authors have already published many different LRI models, why not compare the proposed LRI-CNN with the steerable filters- or power spectrum approach? Furthermore, reporting the computation time for one prediction would be interesting.

In addition, not having a thorough analysis between conventional Conv-layers and the proposed bispectral layer is a missed opportunity. For example, for which image parts do LRI-CNNs outperform the standard U-net and vice versa? Do the feature maps look any different?

**Deanonymize Review:**

no

**Detailed Comments:**

* The 'implementation of the LRI layer' Section is difficult to understand, and a reader trying to reproduce an LRI layer may struggle here. A few sentences explaining n, n', and N would be helpful, including the implications of a large or small N.
* In the text, "i.e" is sometimes used instead of "i.e."
* It seems that some of the Lafarge et al. 2019 citations should be Lafarge et al. 2021.
* Table 1: which N is used here?
* What are the differences between an N=0 LRI-CNN and a standard U-net? Why does the LRI-CNN perform worse?


**Final Rating After The Rebuttal:**

4: Weak Accept

**Justification Of The Final Rating:**

Considering the clarification in the rebuttal and the added results in the final version, I changed my rating to weak accept. Especially the qualitative images and the added forward-pass time may answer in which circumstances the proposed architecture is preferable to standard CNNs.

**Paper Type:**

methodological development

**Questions To Address In The Rebuttal:**

* Why should one prefer a layer that is LRI to a standard convolution layer?
* Are Conv-nets a subset of bispectral layers? I.e., can a bispectral layer model the behavior of a Conv-layer?
* In which scenario should one prefer an RE model with more parameters that performs slightly worse than the baseline regarding the standard F-score?
* Is the LRI-CNN slower than the baseline U-net?
* In the experiments, can the standard U-net be improved by using test-time augmentation?
* What are the advantages and disadvantages of this particular LRI layer compared to other LRI approaches?
* How important is the receptive field size for an LRI layer? Does it, for example, make sense for a layer with a small 5x5 receptive field to be LRI?


**Special Issue:**

no

---

### Meta-Review · Area_Chair_sFcJ · 2022-02-14

**Recommendation:** Accept (Poster)
**Confidence:** 5

**Metareview:**

This paper focuses on designing a novel locally rotation invariant CNN layer. The reviewers raised some initial questions in their comments, which were addressed by the authors in their rebuttal. All reviewers now agree that the paper is ready for publication at MIDL. Based on their recommendation, I'm happy to accept this work.

---

### Decision · Program_Chairs · 2022-02-28

Accept